# Fermented Rubber Seed Kernel with Yeast in the Diets of Tropical Lactating Dairy Cows: Effects on Feed Intake, Hematology, Microbial Protein Synthesis, Milk Yield and Milk Composition

**DOI:** 10.3390/vetsci9070360

**Published:** 2022-07-15

**Authors:** Thanaporn Ouppamong, Nirawan Gunun, Chayapol Tamkhonburee, Pichad Khejornsart, Chatchai Kaewpila, Piyawit Kesorn, Thachawech Kimprasit, Anusorn Cherdthong, Metha Wanapat, Sineenart Polyorach, Suban Foiklang, Pongsatorn Gunun

**Affiliations:** 1Department of Animal Science, Faculty of Natural Resources, Rajamangala University of Technology Isan, Sakon Nakhon Campus, Phangkhon, Sakon Nakhon 47160, Thailand; nuknik007fc@gmail.com (T.O.); chatchai.ka@rmuti.ac.th (C.K.); piyawit.ke@rmuti.ac.th (P.K.); thachawech.ki@rmuti.ac.th (T.K.); 2Department of Animal Science, Faculty of Technology, Udon Thani Rajabhat University, Udon Thani 41000, Thailand; 3Dairy Farming Promotion Organization of Thailand (DPO), Northeast Region, Khon Kaen 40260, Thailand; chayapon.t@dpo.go.th; 4Faculty of Natural Resources and Agro-Industry, Kasetsart University Chalermphrakiat Sakon Nakhon, Province Campus, Sakon Nakhon 47000, Thailand; fnapck@ku.ac.th; 5Tropical Feed Resources Research and Development Center (TROFREC), Department of Animal Science, Faculty of Agriculture, Khon Kaen University, Khon Kaen 40002, Thailand; anusornc@kku.ac.th (A.C.); metha@kku.ac.th (M.W.); 6Department of Animal Production Technology and Fisheries, Faculty of Agricultural Technology, King Mongkut’s Institute of Technology Ladkrabang, Bangkok 10520, Thailand; sineenart.po@kmitl.ac.th; 7Faculty of Animal Science and Technology, Maejo University, Chiang Mai 50290, Thailand; suban@mju.ac.th

**Keywords:** rubber seed kernel, yeast-fermented product, microbial protein synthesis, milk quality, dairy cows

## Abstract

**Simple Summary:**

Rubber seed kernels do not contain sufficient levels of crude protein to serve as a primary source of crude protein in ruminant diets. Yeast has been used to improve the nutritional value of feedstuffs, particularly their protein content, and to diminish their anti-nutritional components. Our earlier studies demonstrated that rubber seed kernels that had undergone yeast fermentation had enhanced levels of crude protein, decreased fiber content, and could be provided to dairy heifers as a source of protein. The current findings showed that the addition of yeast-fermented rubber seed kernels had no negative effects on the hematology, milk production, milk composition, or feed intake. Hence, yeast-fermented rubber seed kernels could be used as a source of protein in the diets of lactating dairy cows.

**Abstract:**

The objective of the present study was to analyze the effects of yeast-fermented rubber seed kernels (YERSEK) on the feed intake, hematology, microbial protein synthesis, milk yield, and milk composition in dairy cows. Six crossbred Holstein Friesian (HF) × Thai lactating dairy cows with 110 ± 10 days in milk were randomly assigned to three different amounts of YERSEK at 0%, 10%, and 20% in a concentrate mixture using a 3 × 3 repeated Latin square design. Cows were fed with concentrate diets at a concentrate-to-milk yield ratio of 1:1.5, with rice straw fed ad libitum. The inclusion of YERSEK did not adversely affect feed intake, nutrient intake, or digestibility (*p* > 0.05), whereas ether extract intake and digestibility linearly increased in dairy cows receiving YERSEK (*p* < 0.01). Increasing YERSEK levels did not adversely affect blood urea nitrogen (BUN) levels, hematological parameters, or microbial protein synthesis (*p* > 0.05). Supplementation of YERSEK did not influence milk production, lactose, or protein levels (*p* > 0.05). However, milk fat and total solids decreased linearly (*p* < 0.05) with the addition of YERSEK. In conclusion, in a concentrate diet, YERSEK could be used as a protein source without negative effects on feed intake, digestibility, hematology, microbial protein synthesis, or milk yield. However, it reduced the milk fat and total solids of tropical lactating dairy cows.

## 1. Introduction

Roughage is often of poor quality in the tropics, with low protein and high fiber content, limiting dairy production efficiency [1,2]. Supplementation of concentrate containing high levels of protein and energy could be useful and diets could be mixed on-farm. Soybean meal (SBM) is a primary protein source in concentrate diets due to its protein-rich and favorable amino acid content [3]. Nevertheless, the cost of cattle feed has been affected by the high cost of SBM [4]. As a result, there is a need to seek alternative dietary protein sources that are less expensive and generated locally to replace SBM in animal diets [5].

A possible alternative dietary protein source is rubber seed (*Hevea brasiliensis*) kernels (RSKs), a by-product of the rubber tree in northeastern Thailand. RSKs contains 19.8–23.6% crude protein (CP), 40.8–47.7% ether extract (EE), and 11.9–20.9% neutral detergent fiber (NDF) [6,7]. However, the protein content of RSKs is not sufficient for use as a main protein source in concentrates in ruminant diets.

Fermentation with bakers’ yeast (*Saccharomyces cerevisiae*) is an easy and cheap method of improving the nutritional value of feed for animal diets. Treatment with yeast and the addition of urea with molasses can increase CP levels and reduce fiber and anti-nutritional factors [8]. Our previous studies revealed that fermentation of RSK with yeast and the addition of urea and molasses improved the CP content by 21.2% to 33.6%, and its use at up to 25% in concentrate did not affect the nutrient intake, nutrient digestion, ruminal metabolism, or microbial protein production of dairy heifers [9]. Pilajun et al. [10] also reported that fermentation of plant meals with yeast and urea with molasses increased CP and non-protein nitrogen (NPN) content.

Due to its quick hydrolysis to NH_3_-N and its absorption from the rumen, the amount of non-protein nitrogen (NPN) that can be employed is limited [11]. Increased blood NH_3_ concentrations increase the risk of ammonia toxicity and related negative health effects in ruminants [12,13]. Disorders of the hematological system and many other organ and systemic diseases can be detected through hematologic analysis [14]. However, no research has been yet carried out on the use of yeast-fermented RSKs in dairy cows. Therefore, the aim of this work was to evaluate the influence of the addition of yeast-fermented rubber seed kernel meal (YERSEK) in concentrate on feed intake, hematology, microbial protein synthesis, milk yield, and milk composition in tropical lactating dairy cows.

## 2. Materials and Methods

### 2.1. Animal Welfare

Animal care was approved by the Animals Ethical Committee of the Rajamangala University of Technology Isan (approval number 10/2564).

### 2.2. Dietary Preparation

From rubber orchards in Sakon Nakhon, Thailand, fresh rubber seeds were harvested. After being hand-picked from the ground, the whole seeds were housed indoors. A dehulling machine was used to remove the seeds’ shells (Incanewlife, Khon Kaen, Thailand). The kernels were crushed through a 1 mm screen after being sun-dried for three days to make YERSEK.

The fermentation solution was produced using a method modified from that of Wanapat et al. [15]. The yeast was activated by mixing 5 g of *S. cerevisiae* with 100 mL of water, and 20 g of sucrose for an hour at room temperature. To create the liquid medium, we combined 42 g of molasses, 100 mL of distilled water, and 40 g of urea. A 1:1 mixture of yeast was added to the liquid medium, and it was air-flushed for 60 h. Two grams of rubber seed kernel was combined with 1 mL of yeast medium, before being air-dried for 72 h. We exposed the samples to sunlight for 48 h, before placing them in a plastic bag and determining their chemical composition.

### 2.3. Animals, Experimental Design, and Treatments

This study was conducted at Kornwadee dairy farm under the administration and supervision of the Dairy Farming Promotion Organization (DPO) in the northeastern region of Thailand. Six mid-lactation crossbred dairy cows (75% HF, 25% Thai native breed) were arranged in a three-by-three replicated Latin square pattern to receive three diets. The cows were 542 ± 30 kg, 110 ± 10 days in milk, and produced 16 ± 1.0 kg of milk each day. Dietary regimens were based on YERSEK levels of 0%, 10%, and 20% in concentrate diets (Table 1). Mid-lactation crossbred dairy cows were fed concentrate diets with a milk production ratio of 1:1.5 [16,17,18] and ad libitum rice straw as roughage. At 7:00 a.m. and 4:00 p.m. each day, diets were provided. The cows were kept in separate enclosures with access to mineral blocks and clean drinking water. Three parts of the investigation, each lasting 21 days, were performed. The first 14 days were spent adapting, whereas the final 7 days were spent collecting data. There was a 7 day transitional period in between each cycle.

### 2.4. Data Collection and Sampling Procedures

Feed and fecal samples were taken from each cow during the final week of the collection period. In the samples, the amounts of dry matter (DM), ash, EE, CP [19], NDF, and acid detergent fiber (ADF) [19,20] were evaluated. Gross energy (GE) was calculated using bomb calorimetry (Oxygen Bomb Calorimeter; Parr Instument Company, Moline, IL, USA). Nutritional digestibility was computed using measurements of the concentration of acid-insoluble ash (AIA) [21]. Over the last five days, urine samples were taken in the morning and the afternoon. Urination was induced by strokes on the side of the vulva. The specimens were examined for allantoin [22] and creatinine [23]. Purine derivative excretions were utilized to gauge the amount of microbial purines consumed and the efficiency of microbial N synthesis (EMNS) based on the relationship obtained by Chen and Gomes [24]. The amount of milk produced each day was recorded. Milk samples were also collected by milking machines during morning and afternoon milking and were refrigerated at 4 °C to determine fat, protein, lactose, total solids, and solids-not-fat by means of infrared methods. Using an automated somatic cell counter, we performed somatic cell counts (SCCs).

Blood samples were collected from the jugular vein 4 h after the morning meal. Blood urea nitrogen (BUN) was determined using Crocker’s technique [25]. Red blood cells (RBC), hemoglobin, hematocrit, mean corpuscular volume (MCV), mean corpuscular hemoglobin (MCH), white blood cells (WBC), neutrophils, lymphocytes, monocytes, and eosinophils were measured using a hematological analyzer (BCC-3000B; DIRUI, Gungoren/Istanbul, Turkey).

### 2.5. Statistical Analysis

The general linear model (GLM) in SAS software was used to analyze the data for variances using a 3 × 3 repeating Latin square design [26]. The dependent variable, Yijk, was evaluated using the model Yijk = μ + Di + Pj + gk + εijk, where Yijk is the dependent variable, μ is the overall mean, Di is the fixed effect of diet, Pj is the fixed effect of the period, gk is the random effect of the cow, and εijk is the residual error. Orthogonal polynomial contrasts were used to assess the treatment means (YERSEK levels) (linear and quadratic). To determine whether an impact was substantial, a threshold of *p* < 0.05 was utilized.

## 3. Results

### 3.1. Chemical Composition of Diets

The concentrate diets were provided with a CP content ranging from 18.3% to 18.6% DM. The CP, EE, NDF, ADF, and GE of RSK were 18.6%, 39.4%, 26.9%, 23.9% DM and 28.0 MJ/kg DM respectively. After the fermentation process, the CP content of YERSEK had increased to 35.7% DM, whereas the EE, NDF, ADF, and GE content had decreased to 32.5%, 24.2%, 15.9% DM and 25.5 MJ/kg DM, respectively (Table 2).

### 3.2. Feed Intake and Nutrient Digestibility

Increasing levels of YERSEK did not influence the feed intake (*p* > 0.05) (Table 3). Nutrient intake and digestibility were similar among groups (*p* > 0.05), whereas EE intake and digestibility increased linearly (*p* < 0.01) with increasing levels of YERSEK.

### 3.3. Blood Urea Nitrogen and Hematological Parameters

The inclusion of YERSEK did not affect the levels of BUN, RBC, WBC, MCV, MCH, lymphocytes, neutrophils, monocytes, eosinophils, hemoglobin, or hematocrit (*p* > 0.05) (Table 4).

### 3.4. Microbial Protein Synthesis

The inclusion of YERSEK did not have a significant effect on the urinary purine derivatives, microbial CP synthesis, or EMNS (*p* > 0.05) (Table 5).

### 3.5. Milk Production and Compositions

Increasing levels of YERSEK did not affect milk production (*p* > 0.05) (Table 6). The use of YERSEK in concentrate feed for dairy cows had also no effect on milk protein, lactose, or solids-not-fat (*p* > 0.05), but milk fat and total solids linearly decreased (*p* < 0.05). The SCCs were similar among treatments (*p* > 0.05). 

## 4. Discussion

### 4.1. Chemical Composition of Diets

The CP content of YERSEK increased from 18.6% DM to 35.7% DM. This result is in agreement with our previous studies, which showed that the CP content of YERSEK increased from 21.2% DM to 33.6% DM [9]. This result could be due to the supplementation of urea and molasses as nitrogen and carbon sources, respectively. The reduced EE of YERSEK compared with RSK suggests that plant oil can be degraded into hydroperoxides under the effect of light and oxygen during solid-state fermentation [27]. This radical can generate peroxyl radicals by taking a hydrogen atom from the lipid or a hydrogen atom from lipid hydroperoxides [28]. In addition, the reduced NDF and ADF contents of YERSEK could result from two plausible processes. First, the fiber content of the substances used to transform RSK into YERSEK, such as yeast, sucrose, molasses, and urea, may have decreased, which caused the NDF and ADF contents in YERSEK to be diluted [9]. Second, the fiber content of YERSEK may have been degraded due to the secretion of different enzymes by yeasts and naturally occurring cellulase-producing microorganisms [29]. Gunun et al. [9] also reported a decrease in EE, NDF, and ADF contents in YERSEK with the addition of yeast, sucrose, and molasses during solid-state fermentation.

### 4.2. Feed Intake and Nutrient Digestibility

It has frequently been noted that the lower palatability of supplemental fat to ruminants may be the cause of the negative impact of fat on intake [6,30,31]. Our earlier research revealed that rice straw intake, total intake, and fiber intake decreased when dairy heifers were fed YERSEK at 15–25% in concentrate diets [9]. In contrast, the addition of YERSEK in concentrate did not affect the feed intake or fiber intake of dairy cows in the present study, which could indicate suitable levels for rumen microbial activity and also nutrient intake [32,33,34]. However, the inclusion of YERSEK in the diet could increase the intake and digestibility of EE. This is consistent with the results of our previous work, in which we included YERSEK in concentrate to improve EE intake and digestibility in dairy heifers [9]. These results might be due to high EE content in the diet, which enhanced EE intake and digestibility. Moreover, another study reported a decrease in NDF and ADF digestibility using RSK in goats at higher levels (30%) in concentrate [6]. This effect may be due to unsaturated fatty acids, which are toxic to ruminal bacteria, especially cellulolytic bacteria [35]. In contrast, fiber digestibility was not affected by the addition of YERSEK in the current study. Similarly, Gunun et al. [9] reported that the inclusion of YERSEK at 5–25% in the diet had no effect on the fiber digestibility in dairy heifers. These results suggest that dairy cows can utilize up to 20% YERSEK in the diet with no effect on nutrient intake or digestibility.

### 4.3. Blood Urea Nitrogen and Hematological Parameters

The BUN concentration is widely used to monitor protein status and metabolic conditions related to health and diseases [36]. In our study, the addition of YERSEK did not influence the BUN concentration, which is consistent with the results of Gunun et al. [9], who reported that the inclusion of YERSEK had no effect on BUN concentrations at 4 h post-feeding in dairy heifers. Hematological indices are related to nutritional status. They have been used to monitor and evaluate animal health and nutritional status, and the cause of an abnormality or dysfunction in ruminants is often assessed through blood analysis [37]. The inclusion of YERSEK did not affect hemogram indicators. Similarly, Piamphon et al. [38] noted that hemoglobin and hematocrit were unaffected when feeding yeast-fermented Napier grass mixed with cassava root to beef cattle. In comparison to the previous findings, all of the hematological indices for ruminants remained within their respective ranges [17,39,40]. Based on these findings, it can be assumed the addition of YERSEK at any level to the diet of dairy cows is non-toxic.

### 4.4. Microbial Protein Synthesis

Microbial CP is important for ruminant protein supply and provides the majority of amino acids required for animal activity and growth [41]. The amount of microbial CP flow from the rumen, which is a result of microbial growth and its outflow from the rumen, is one measure of the effectiveness of rumen nitrogen utilization [42]. The effects of YERSEK on PD and microbial CP synthesis in ruminants have been investigated in limited studies. Gunun et al. [9] reported that the inclusion of YERSEK in concentrate had no effect on PD or microbial CP synthesis in dairy heifers. Similarly, PD and microbial CP synthesis were not affected by the inclusion of YERSEK in the present study. When beef cattle were fed yeast-fermented cassava root, Promkot and Pornanek [43] reported that this had no effect on the synthesis of microbial proteins in the rumen. Moreover, EMNS was unaffected by YERSEK supplementation, which is consistent with our earlier findings that the inclusion of YERSEK in concentrate had no effect on EMNS in dairy heifers [9]. These results suggest that YERSEK did not enhance rumen ammonia or branched-chain amino acid levels; hence, it did not affect microbial CP in the rumen.

### 4.5. Milk Production and Composition

Milk yield was not affected, but milk fat and total solid contents were reduced in dairy cows fed YERSEK in concentrate. In contrast, Udo et al. [44] reported that milk yield and milk composition were unaffected by the inclusion of boiled rubber seed kernel at 10–30% in the concentrate of West African Dwarf goats. The unsaturated fatty acid content of the rubber seed kernel was 80.8%, mainly including 37.8% linoleic acid (C18:2) and 17.6% linolenic acid (C18:3) [7]. The inclusion of unsaturated fatty acids in the diet can increase *trans*-fatty acid production in the rumen due to incomplete biohydrogenation. *Trans*-fatty acids, particularly *trans*-10 and *cis*-12 conjugated linoleic acid, inhibit the de novo synthesis of fatty acids in milk through reducing the expression of lipogenic enzymes (e.g., acetyl-CoA carboxylase and fatty acid synthase) in the mammary gland [45,46]. These mechanisms revealed that the inclusion of YERSEK in concentrate decreased milk fat content in dairy cows.

## 5. Conclusions

Feed with the inclusion of YERSEK at 10–20% in concentrate was able to be used as a protein source and this had no effect on feed utilization, hematological parameters, microbial protein synthesis, or milk production, whereas milk fat and total solids decreased in tropical lactating dairy cows.

## Figures and Tables

**Table 1 vetsci-09-00360-t001:** Ingredients of the diet used in the experiment.

Item	Level of YERSEK (%DM)		
	0	10	20
Ingredient, kg dry matter (DM)			
Cassava chip	53.0	53.0	53.0
Soybean meal	27.0	20.0	13.3
YERSEK	0.0	10.0	20.0
Coconut meal	8.4	6.9	5.1
Rice bran	6.0	4.5	3.0
Molasses	2.5	2.5	2.5
Urea	1.1	1.1	1.1
Mineral and vitamin mixture	1.0	1.0	1.0
Salt	0.5	0.5	0.5
Sulfur	0.5	0.5	0.5

RSK, rubber seed kernel; YERSEK, yeast-fermented rubber seed kernel.

**Table 2 vetsci-09-00360-t002:** Chemical composition of concentrate, rice straw, RSK, and YERSEK.

Item	Level of YERSEK (%DM)			Rice Straw	RSK	YERSEK
	0	10	20			
Chemical composition						
Dry matter, %	88.8	88.4	88.7	95.3	94.9	97.5
Organic matter, %DM	94.3	94.6	94.5	90.5	96.7	96.6
Crude protein, %DM	18.2	18.3	18.6	2.6	18.6	35.7
Ether extract, %DM	1.3	5.4	9.8	0.6	39.4	32.5
Neutral detergent fiber, %DM	28.7	27.6	29.3	74.9	26.9	24.2
Acid detergent fiber, %DM	14.8	16.7	15.8	54.7	23.9	15.9
Ash, %DM	5.7	5.4	5.5	9.5	3.4	3.4
Gross energy, MJ/kg DM	20.5	25.0	26.3	13.2	28.0	25.5
Price, Thai baht/kg	11.1	10.7	10.4	-	-	-

RSK, rubber seed kernel; YERSEK, yeast-fermented rubber seed kernel.

**Table 3 vetsci-09-00360-t003:** Effect of yeast-fermented rubber seed kernel (YERSEK) on intake and nutrient digestibility in dairy cows.

Item	Level of YERSEK (%DM)			SEM	Contrast	
	0	10	20		*p*-Linear	*p*-Quadratic
DM intake, kg/d						
Rice straw	5.4	5.2	4.8	0.09	0.09	0.63
Concentrate	9.9	10.1	10.2	0.12	0.56	0.82
Total intake	15.3	15.4	15.0	0.14	0.51	0.62
Nutrient intake, kg/d						
Organic matter	14.2	14.3	14.0	0.18	0.57	0.60
Crude protein	2.0	2.0	2.0	0.03	0.37	0.90
Ether extract	0.2	0.6	1.0	0.04	<0.01	0.79
Neutral detergent fiber	7.4	7.2	7.0	0.18	0.22	0.96
Acid detergent fiber	4.4	4.5	4.2	0.08	0.31	0.16
Digestibility coefficients, %						
Dry matter	71.9	69.4	68.5	1.03	0.14	0.66
Organic matter	74.2	71.6	70.6	1.34	0.12	0.65
Crude protein	71.5	69.1	67.2	1.51	0.11	0.92
Ether extract	75.4	91.4	93.5	0.67	<0.01	0.00
Neutral detergent fiber	57.5	55.2	56.4	0.82	0.71	0.50
Acid detergent fiber	54.8	51.3	50.3	1.73	0.19	0.66

**Table 4 vetsci-09-00360-t004:** Effect of yeast-fermented rubber seed kernel (YERSEK) on blood urea nitrogen (BUN) and hematological parameters in dairy cows.

Item	Level of YERSEK (%DM)			SEM	Contrast	
	0	10	20		*p*-Linear	*p*-Quadratic
Blood urea nitrogen, mg/dL	15.3	17.5	13.6	1.34	0.84	0.08
Red blood cell, 10^12^/L	5.0	4.7	5.5	0.14	0.39	0.28
Hemoglobin, g/dL	8.6	8.1	9.4	0.56	0.43	0.36
Hematocrit, %	25.9	24.6	28.3	1.39	0.46	0.37
Mean corpuscular volume, fL	52.9	53.3	53.5	0.49	0.75	0.93
Mean corpuscular hemoglobin, pg	17.8	17.1	18.3	0.48	0.72	0.44
White blood cells, 10^9^/L	9.0	8.9	11.3	0.85	0.24	0.42
Neutrophils, %	68.8	69.8	64.8	1.84	0.29	0.42
Lymphocytes, %	28.9	27.5	34.3	1.88	0.44	0.49
Monocytes, %	0.9	0.4	0.7	0.20	0.69	0.11
Eosinophils, %	1.6	2.2	3.2	0.62	0.23	0.53

**Table 5 vetsci-09-00360-t005:** Effect of yeast-fermented rubber seed kernel (YERSEK) on microbial protein synthesis in dairy cows.

Item	Level of YERSEK (%DM)			SEM	Contrast	
	0	10	20		*p*-Linear	*p*-Quadratic
Urinary purine derivatives (mmol/d)						
Purine excretion	98.2	104.2	111.6	8.41	0.30	0.94
Purine absorption	79.3	87.3	97.2	11.08	0.28	0.95
Urine creatinine	5.9	5.7	6.7	0.33	0.81	0.40
MN (g/d)	57.6	63.5	70.6	8.06	0.28	0.95
MCP (g/d)	360.2	396.7	441.7	50.35	0.28	0.95
EMNS (g/kg OMDR)	7.7	10.1	10.2	1.19	0.24	0.73

MN, microbial nitrogen; MCP, microbial crude protein; EMNS, efficiency of microbial N synthesis; OMDR, organic matter digested in the rumen.

**Table 6 vetsci-09-00360-t006:** Effect of yeast fermented rubber seed kernel (YERSEK) on milk yield and milk composition in dairy cows.

Item	Level of YERSEK (%DM)			SEM	Contrast	
	0	10	20		*p*-Linear	*p*-Quadratic
Production, kg/day						
Milk yield	16.7	17.2	16.8	0.15	0.86	0.33
4% FCM	15.9	14.8	13.5	1.07	0.06	0.96
Milk composition, %						
Fat	3.8	3.4	2.9	0.09	0.01	0.90
Protein	3.3	3.2	3.1	0.13	0.10	0.49
Lactose	4.5	4.6	4.6	0.05	0.09	0.29
Solids-not-fat	8.7	8.8	8.8	0.04	0.73	0.82
Total solids	12.6	12.3	12.0	0.28	0.04	0.94
Somatic cell counts (*n*/mL) 10^5^	2.3	3.9	1.3	1.3	0.65	0.30

FCM, fat-corrected milk.

## Data Availability

Not applicable.

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
