# Peer review of "Fermented Rubber Seed Kernel with Yeast in the Diets of Tropical Lactating Dairy Cows: Effects on Feed Intake, Hematology, Microbial Protein Synthesis, Milk Yield and Milk Composition"

_vetsci, 2022, doi:10.3390/vetsci9070360_

Round 1
Reviewer 1 Report
It is and interesting paper that offers information about the use of a byproduct in dairy cattle feeding.
But of the three more important issues relative to the use of yersek, palatability, safety and economical interest, the last one must be better explained, especially in the relative to how milk components are paid.
Typo errors:
Line 64: Please, change digstion to digestion
Line 65: Please, change microbe to microbial
Line 94: Please correct the daily milk output (16 1.0 kg/day??)
Minor corrections:
Line 27: Why feed intakes appear as first objective in the abstract and do not appear in the title?
Line 28: change Holstein by crossbred Holstein as you said in the line 93
Lines 36 and 37: It is contradictory to write in the same phrase that YERSEK supplementation not interfere milk composition and that fat in milk decreased linearly. If supplementation decreases fat milk then interfere with milk composition. Please rewrite the sentence.
Lines 36 to 41: The last sentence –Therefore, the use of Yersek at 10%...- cannot be explained by the previous sentences. Please rewrite the sentence and the Conclusions
Line 86: Please, specify the type of sugar
Lines 89 and 90: how many time acts the yeast medium on the RSK before to be air dried?
Line 111: What method was used for take the urine sampling?
Lines 131 and 132: Can you explain what do you mean when you said: -The improved nutrient composition of RSK by fermentation with yeast are important 131 when the feedstuff is new.- Sorry but I am not able to understand it.
Lines 157 and 158: you write: -Moreover, feed cost and margin over feed were not influenced by dietary treatments (p > 0.05).- for calculate the margin over feed, that appear in the table 5, you must include in Materials and Methods how do you calculate it. You must explain how milk is paid, just for liters or if fat and somatic cell count (SCC) also are considered. As you show in table 5 there are important differences in fat milk and SCC among groups and in many countries this imply different milk price.
Lines 224 to 226: Please erase from the sentence -or milk composition- because the next sentence explain that milk fat decreases.
Lines 227 and 228: The sentence -The optimum level of 10% YERSEK in concentrate can lower feed cost in lactating dairy cows fed on rice straw.- has not been explain in Discussion
Author Response
It is and interesting paper that offers information about the use of a byproduct in dairy cattle feeding.
-Thank you very much for your interest.
But of the three more important issues relative to the use of yersek, palatability, safety and economical interest, the last one must be better explained, especially in the relative to how milk components are paid.
-I’ve removed the economic return in Table 6, followed by the recommendation of reviewer 3.
Typo errors:
Line 64: Please, change digstion to digestion
-Line 76: Already changed.
Line 65: Please, change microbe to microbial
-Line 77: Already changed.
Line 94: Please correct the daily milk output (16 1.0 kg/day??)
-Line 107: Already changed to “16±1.0 kg of milk each day”, please see in text.
Minor corrections:
Line 27: Why feed intakes appear as first objective in the abstract and do not appear in the title?
-Already changed title to “Fermented rubber seed kernel with yeast in the diets of tropical lactating dairy cows: Effect on feed intake, hematology, microbial protein synthesis, milk yield and milk composition.”, please see in text.
Line 28: change Holstein by crossbred Holstein as you said in the line 93
-Line 39: Already changed to “Six crossbred Holstein Friesian”, please see in text.
Lines 36 and 37: It is contradictory to write in the same phrase that YERSEK supplementation not interfere milk composition and that fat in milk decreased linearly. If supplementation decreases fat milk then interfere with milk composition. Please rewrite the sentence.
Line 47-50: Already changed to “Supplementation of YERSEK did not influence milk production, lactose and protein (p > 0.05). However, milk fat and total solid decreased linearly
(p < 0.05) by addition of YERSEK at 10% in concentrate.”, please see in text.
Lines 36 to 41: The last sentence –Therefore, the use of Yersek at 10%...- cannot be explained by the previous sentences. Please rewrite the sentence and the Conclusions
-Line 50 to 54: Already changed to “In conclusion, the inclusion of YERSEK at 20% in concentrate affects milk fat and total solid. Therefore, the use of YERSEK at 10% in concentrate diet could be used as a protein source without negative effects on feed intake, digestibility, hematology, microbial protein synthesis, milk yield and milk composition for tropical lactating dairy cows.”, please see in text.
Line 86: Please, specify the type of sugar
-Line 98-99: Already changed to “20 g of sucrose”, please see in text.
Lines 89 and 90: how many time acts the yeast medium on the RSK before to be air dried?
-Line 101-102: Already changed to “2 g of rubber seed kernel was combined with 1 ml of yeast medium before being air-dried for 72 hours.”, please see in text.
Line 111: What method was used for take the urine sampling?
-Line 125-126: Already changed to “Over the last five days, urine samples were taken in the morning and the afternoon. Strokes on the vulva's side caused the patient to urinate.”, please see in text.
Lines 131 and 132: Can you explain what do you mean when you said: -The improved nutrient composition of RSK by fermentation with yeast are important 131 when the feedstuff is new.- Sorry but I am not able to understand it.
-Already deleted, please see in text.
Lines 157 and 158: you write: -Moreover, feed cost and margin over feed were not influenced by dietary treatments (p > 0.05).- for calculate the margin over feed, that appear in the table 5, you must include in Materials and Methods how do you calculate it. You must explain how milk is paid, just for liters or if fat and somatic cell count (SCC) also are considered. As you show in table 5 there are important differences in fat milk and SCC among groups and in many countries this imply different milk price.
-I’ve removed the economic return in Table 6, followed by the recommendation of reviewer 3.
Lines 224 to 226: Please erase from the sentence -or milk composition- because the next sentence explain that milk fat decreases.
-Line 265-269: Already changed to “The inclusion of YERSEK at 10%–20% in concentrate had no effect on feed utilization, hematological parameters, microbial protein synthesis and milk production of lactating dairy cows. However, milk fat and total solid content was decreased with 20% YERSEK in concentrate. The optimum level of 10% YERSEK in concentrate in tropical lactating dairy cows fed on rice straw.”, please see intext.
Lines 227 and 228: The sentence -The optimum level of 10% YERSEK in concentrate can lower feed cost in lactating dairy cows fed on rice straw.- has not been explain in Discussion
-Already deleted feed cost in table 6 and text.
Reviewer 2 Report
Manuscript vetsci-1777663, entitled “Fermented rubber seed kernel with yeast in the diets of tropical lactating dairy cows: Effects on hematology, microbial protein synthesis, milk yield and milk composition”
Recommendation: The above paper is not suitable for publication in its present form.
General comment
The article provides useful information about the effects of fermented rubber seed kernel with yeast dietary supplementation on hematology, microbial protein synthesis, milk yield and milk composition in tropical lactating dairy cows. Although, the experiment is in general appropriately designed and implemented, there are some points that should be corrected or clarified.
Major comments
1) Too small sample size for assessing the effects on feed intake, milk yield and milk composition.
2) How was the quantity of the offered feed calculated based on milk yield (1:1.5)? NRC or AFRC?
3) Please add P-values in Tables, apart from P-linear and P-quadratic
4) Inadequate discussion. Authors mainly compare the results of the present study with their previous unpublished data.
5) How was feed cost calculated and analyzed? Please provide details. At the same time, authors conclude that feed cost could be reduced by YERSEK supplementation. However, according to Table 5, feed cost was not affected.
Minor points:
L26: “The objective of the present study was to…”
L36: “influence” instead of “interfere”
L49-50: “…diets due to its protein-rich…”
L51: “seek on” instead of “look into”
L64: “nutrient digestion”
L68: “Due to the” instead of “Because of”
L72: “yet carried out” instead of “done”
L74: “…meal (YERSEK) addition in concentrate…”
L74: “…milk yield and milk composition in tropical…”
L84: “produced” instead of “conducted”
L85-88: Please rephrase
L94: “161.0 kg/day”? Please check
L98: “Diets were provided” instead of “It was served to all diets”
L115: “…was recorded. Milk samples were also collected…”
L117: “collected” instead of “taken”
L135: Please provide CP value
L141: “influence” instead of “change the”
L147: “…affect the levels of BUN…”
L151: “have a significant effect on” instead of “change”
L155: “…had also no effect on…”
L169-170: Please rephrase. What do you mean?
L186: “reported” instead of “noted”
L189: Dairy heifers or cows?
L207: 49.3%? Where is this number presented?
L221: “mechanisms” instead of “results”
Author Response
General comment
The article provides useful information about the effects of fermented rubber seed kernel with yeast dietary supplementation on hematology, microbial protein synthesis, milk yield and milk composition in tropical lactating dairy cows. Although, the experiment is in general appropriately designed and implemented, there are some points that should be corrected or clarified.
Major comments
1) Too small sample size for assessing the effects on feed intake, milk yield and milk composition.
- I’m agree with small sample size for evaluate in dairy cows. However, we are using 6 lactating dairy cows experiment in smallholder dairy farming in Thailand. Many experiment use for evaluate for feed intake, milk yield and milk composition such as:
Insoongnern, H.; Srakaew, W.; Prapaiwong, T.; Suphrap, N.; Potirahong, S.;Wachirapakorn, C. 2021. Effect of mineral salt blocks containing sodium bicarbonate or selenium on ruminal pH, rumen fermentation and milk production and composition in crossbred dairy cows. Vet. Sci. 8: 322. They use 4 dairy cows.
Gunun, P., N. Gunun, P. Khehornsart, T. Ouppamong, A. Cherdthong, M. Wanapat, S. Sililaophaisan, C. Yuangklang, S. Polyorach, W. Kenchaiwong, and S. Kang. 2019. Effects of Antidesma thwaitesianum Muell. Arg. Pomace as a source of plant secondary compounds on digestibility, rumen environment, hematology, and milk production in dairy cows. Anim. Sci. J. 90:372-381. They use 4 dairy cows.
Prachumchai, R., A. Cherdthong, M. Wanapat, S. So and S. Polyorach. 2022. Fresh cassava root replacing cassava chip could enhance milk production of lactating dairy cows fed diets based on high sulfur-containing pellet. Scientific reports. 12:3809. They use 4 dairy cows.
Benchaar, C., H.V. Petit, R. Berthiaume, and T.D. Whyte. 2006. Effects of Addition of Essential Oils and Monensin Premix on Digestion, Ruminal Fermentation, Milk Production, and Milk Composition in Dairy Cows. J. Dairy Sci. 89:4352-4364. They use 4 dairy cows.
2) How was the quantity of the offered feed calculated based on milk yield (1:1.5)? NRC or AFRC?
-Line 111: We are feeding mid-lactation Thai crossbred dairy cows in our experiment (feed:milk yield 1:1.5) according to:
Phesatcha, K.; Wanapat, M. Performance of lactating dairy cows fed a diet based on treated rice straw and supplemented with pelleted sweet potato vines. Trop. Anim. Health Prod. 2013, 45, 533-538.
Gunun, P.; Gunun, N.; Khejornsart, P.; Ouppamong, T.; Cherdthong, A.; Wanapat, M.; Sililaophaisan, S.; Yuangklang, C.; Polyorach, S.; Kenchaiwong, W.; Kang, S. Effects of Antidesma thwaitesianum Muell. Arg. pomace as a source of plant secondary compounds on digestibility, rumen environment, hematology, and milk production in dairy cows. Anim. Sci. J. 2019, 90, 372-381.
Insoongnern, H.; Srakaew, W.; Prapaiwong, T.; Suphrap, N.; Potirahong, S.; Wachirapakorn, C. Effect of mineral salt blocks containing sodium bicarbonate or selenium on ruminal pH, rumen fermentation and milk production and composition in crossbred dairy cows. Vet. Sci. 2021, 8, 322.
, please see in text.
3) Please add P-values in Tables, apart from P-linear and P-quadratic.
-Already add P-value (P-linear and P-quadratic), please see in Table 3,4,5 and 6.
4) Inadequate discussion. Authors mainly compare the results of the present study with their previous unpublished data.
-L182-263: Already improved discussion, please see in text.
5) How was feed cost calculated and analyzed? Please provide details. At the same time, authors conclude that feed cost could be reduced by YERSEK supplementation. However, according to Table 5, feed cost was not affected.
-I’ve deleted the economic return and feed cost in Table 6, followed by the recommendation of reviewer 3.
Minor points:
L26: “The objective of the present study was to…”
-L37: Already changed, please see in text.
L36: “influence” instead of “interfere”
-L48: Already changed, please see in text.
L49-50: “…diets due to its protein-rich…”
-L62-63: Already change, please see in text.
L51: “seek on” instead of “look into”
-L64: Already change, please see in text.
L64: “nutrient digestion”
-L77: Already change, please see in text.
L68: “Due to the” instead of “Because of”
-L81: Already change, please see in text.
L72: “yet carried out” instead of “done”
-L86: Already change, please see in text.
L74: “…meal (YERSEK) addition in concentrate…”
-L87-88: Already change, please see in text.
L74: “…milk yield and milk composition in tropical…”
-L89: Already change, please see in text.
L84: “produced” instead of “conducted”
-L97: Already change, please see in text.
L85-88: Please rephrase
L98-104: Already changed to “The yeast was activated by mixing 5 g of S. cerevisiae with 100 ml of water, 20 g of sucrose, and an hour at room temperature. Urea, molasses, and water were combined in a ratio of 40:42:100 to create the liquid medium. A 1:1 mixture of yeast was added to the liquid medium, and it was air-flushed for 60 hours. 2 g of rubber seed kernel was combined with 1 ml of yeast medium before being air-dried for 72 hours. The samples were exposed to sunlight for 48 hours before being placed in a plastic bag and having their chemical makeup examined.”, please see intext.
L94: “161.0 kg/day”? Please check
-L107: Already changed to “16±1.0 kg of milk each day”, please see in text.
L98: “Diets were provided” instead of “It was served to all diets”
-L112: Already changed, please see in text.
L115: “…was recorded. Milk samples were also collected…”
-L130: Already changed, please see in text.
L117: “collected” instead of “taken”
-L133: Already changed, please see in text.
L135: Please provide CP value
-L151-152: Already changed to “After the fermented process, the CP content of YERSEK was increased to 35.7 %DM”, please see in text.
L141: “influence” instead of “change the”
-L151: Already changed, please see in text.
L147: “…affect the levels of BUN…”
-L163: Already changed, please see in text.
L151: “have a significant effect on” instead of “change”
-L169: Already changed, please see in text.
L155: “…had also no effect on…”
-L176: Already changed, please see in text.
L169-170: Please rephrase. What do you mean?
-Already deleted, please see in text.
L186: “reported” instead of “noted”
-L212: Already changed, please see in text.
L189: Dairy heifers or cows?
-L218: Already changed to “dairy cows”, please see in text.
L207: 49.3%? Where is this number presented?
-Already deleted this sentence, please see in text.
L221: “mechanisms” instead of “results”
-L261: Already changed, please see in text.
Reviewer 3 Report
The paper entitled “Yeast-fermented gum seed kernel in the diet of tropical lactating dairy cows: Effects on hematology, microbial protein synthesis, milk yield and milk composition”, is a field study with the objective of analysing the effects of yeast-fermented gum seed kernel (YERSEK) on feed intake, hematology, microbial protein synthesis, milk yield and milk composition in dairy cows (crossbred HF x Thai).
The paper has limitations, including the presence in the text of many “our unpublished, unpublished data with the author's surname Gunun et al;”, materials and methods need to be expanded, Table 1 needs to be improved, and the economic approach on costs as well as being incorrect needs to be removed from table 5 and text.
The paper to be published needs to be greatly improved by incorporating the following points listed below:
line 28: rewrite “crossbreed HF x Thai”
Line 112-117: It is necessary to better explain this paragraph in particular the milking system used to milk the animals, the method used to measure individual milk yield, the methods used to determine milk quality (fat, protein ...) and somatic cells.
Line 124-127: specify the statistical SW used for statistical analysis and expression of results.
Line 137: restructure table 1 to make it more understandable.
Line 154-155: “Increasing levels of YERSEK did not affect milk production (p > 0.05) (Table 5). The use of YERSEK in concentrate feed for dairy cows had no effect on milk protein, lactose, total solids, and solids-not-fat (p > 0.05), but milk fat was linearly decreased (p < 0.05). The parameter 'total solid' presents a significance level p=0.04 (table 5), correct the considerations in the text and in the conclusion.
Line 179: Table 5, should be included in the “results section”. Rewrite “ Somatic cell count (n/ml) 10^5 “
Line 223-228: the conclusions should be reworded, removing the economic considerations. I also suggest that the Authors, for the future, evaluate costs by reference to normalised milk at 4% FCM
.Check the following references: 15, 18, 19, 20, 22, 29,
Author Response
The paper has limitations, including the presence in the text of many “our unpublished, unpublished data with the author's surname Gunun et al;”, materials and methods need to be expanded, Table 1 needs to be improved, and the economic approach on costs as well as being incorrect needs to be removed from table 5 and text.
- The presence in the text of many “our unpublished, unpublished data with our previous studies, now published in FERMENTATION and changed in our manuscript “Gunun et al. (2022)
Gunun, N.; Ouppamong, T.; Khejornsart, P.; Cherdthong, A.; Wanapat, M.; Polyorach, S.; Kaewpila, C.; Kang, S.; Gunun, P. Effects of rubber seed kernel fermented with yeast on feed utilization, rumen fermentation and microbial protein synthesis in dairy heifers. Fermentation 2022, 8, 288.
The paper to be published needs to be greatly improved by incorporating the following points listed below:
Line 28: rewrite “crossbreed HF x Thai”
-L39: Already changed to “crossbred Holstein Friesian (HF) × Thai”, please see in text.
Line 112-117: It is necessary to better explain this paragraph in particular the milking system used to milk the animals, the method used to measure individual milk yield, the methods used to determine milk quality (fat, protein ...) and somatic cells.
-Line 130-133: Already changed to “Milk samples were also collected during morning and afternoon milking and refrigerated at 4°C to determine fat, protein, lactose, total solids, and solids-not-fat by infrared methods. Using an automated somatic cell counter, do a somatic cell count (SCC).”, please see in text.
Line 124-127: specify the statistical SW used for statistical analysis and expression of results.
-Line 141-147: Already changed to “The general linear model (GLM) in SAS software was used to analyze the data for variances using a 3×3 repeating Latin square design [26]. The dependent variable, Yijk, was evaluated using the model Yijk = μ + Di + Pj + gk + eijk, where Yijk is the dependent variable, μ is the overall mean, Di is indeed the fixed effect of diet, Pj is the fixed effect of period, gk is the random effect of cow, and eijk is the residual error. Orthogonal polynomial contrasts were used to assess the treatment means (YERSEK levels) (linear and quadratic). To determine if an impact was substantial, p < 0.05 was utilized.
Line 137: restructure table 1 to make it more understandable.
-Table 1 and 2: Already divide Table 1 into Table 1 and Table 2.
Table 1. Ingredients of the diet used in the experiment.
Item |
Level of YERSEK (%DM) |
||
0 |
10 |
20 |
|
Ingredient, kg dry matter (DM) |
|
|
|
Cassava chip |
53.0 |
53.0 |
53.0 |
Soybean meal |
27.0 |
20.0 |
13.3 |
YERSEK |
0.0 |
10.0 |
20.0 |
Coconut meal |
8.4 |
6.9 |
5.1 |
Rice bran |
6.0 |
4.5 |
3.0 |
Molasses |
2.5 |
2.5 |
2.5 |
Urea |
1.1 |
1.1 |
1.1 |
Mineral and vitamin mixture |
1.0 |
1.0 |
1.0 |
Salt |
0.5 |
0.5 |
0.5 |
Sulfur |
0.5 |
0.5 |
0.5 |
RSK, rubber seed kernel; YERSEK, yeast fermented rubber seed kernel.
Table 2. Chemical composition of concentrate, rice straw, RSK and YERSEK.
Item |
Level of YERSEK (%DM) |
Rice straw |
RSK |
YERSEK |
||
0 |
10 |
20 |
||||
Chemical composition |
|
|
|
|
|
|
Dry matter, % |
88.8 |
88.4 |
88.7 |
95.3 |
94.9 |
97.5 |
Organic Matter, %DM |
94.3 |
94.6 |
94.5 |
90.5 |
96.7 |
96.6 |
Crude Protein, %DM |
18.2 |
18.3 |
18.6 |
2.6 |
18.6 |
35.7 |
Ether extract, %DM |
1.3 |
5.4 |
9.8 |
0.6 |
39.4 |
32.5 |
Neutral detergent fiber, %DM |
28.7 |
27.6 |
29.3 |
74.9 |
26.9 |
24.2 |
Acid detergent fiber, %DM |
14.8 |
16.7 |
15.8 |
54.7 |
23.9 |
15.9 |
Ash, %DM |
5.7 |
5.4 |
5.5 |
9.5 |
3.4 |
3.4 |
Gross Energy, MJ/kg DM |
20.5 |
25.0 |
26.3 |
13.2 |
28.0 |
25.5 |
Price, Thai baht/kg |
11.1 |
10.7 |
10.4 |
- |
- |
- |
RSK, rubber seed kernel; YERSEK, yeast fermented rubber seed kernel.
Line 154-155: “Increasing levels of YERSEK did not affect milk production (p > 0.05) (Table 5). The use of YERSEK in concentrate feed for dairy cows had no effect on milk protein, lactose, total solids, and solids-not-fat (p > 0.05), but milk fat was linearly decreased (p < 0.05). The parameter 'total solid' presents a significance level p=0.04 (table 5), correct the considerations in the text and in the conclusion.
-L176-179: Already changed to “Increasing levels of YERSEK did not affect milk production (p > 0.05) (Table 6). The use of YERSEK in concentrate feed for dairy cows had also no effect on milk protein, lactose, and solids-not-fat (p > 0.05), but milk fat and total solid were linearly decreased (p < 0.05).”, please see in text.
Line 179: Table 5, should be included in the “results section”. Rewrite “ Somatic cell count (n/ml) 10^5 “
-Line 179 and Table 6: Already included and changed “Somatic cell count (n/ml) 10^5”, please see in text and table.
Line 223-228: the conclusions should be reworded, removing the economic considerations. I also suggest that the Authors, for the future, evaluate costs by reference to normalised milk at 4% FCM
-Already removed the economic return in Table 6.
Check the following references: 15, 18, 19, 20, 22, 29,
-Already checked and changed references 19 to
- Hawk, P.B.; Oser, B.L.; Summerson, W.H. Practical Physiological Chemistry, 13th ed.; McGraw Hill Publishing Company Ltd., London, UK, 1954.
,please see in text.
Round 2
Reviewer 2 Report
Authors made the majority of the necessary amendments. However, some points should be corrected before the acceptance of their article
L26-27: “Rubber seed kernel does not contain sufficient levels of crude protein to serve as its primary source in ruminant diets.”
L30: “…enhanced levels of crude protein…”
L31: “provided” instead of “fed”
L31-32: Please delete “The impact of hematological, microbial protein synthesis, milk production, and milk quality in dairy cows is undefined.”
L34: “feed intake” instead of “the consumption of feed”
L35-36: “Hence, yeast-fermented rubber seed kernel could be used as a source of protein in diets of lactating dairy cows.”
L49-50: Please delete “at 10% in concentrate”
L50-54: This conclusion is not supported by your findings. P-linear shows a linear trend after increasing YERSEK levels. As shown, the level of both 10 and 20% resulted in a decrease of milk fat and total solid.
L89: “yield”
L98-99: “The yeast was activated by mixing 5 g of S. cerevisiae with 100 ml of water, and 20 g of sucrose for an hour at room temperature.”
L103-104: “and determining their chemical composition” instead of “and having their chemical 103 makeup examined”
L126: “Patient”?
L203: “contrast” instead of “disagreement with the current study”
L205: “…of dairy cows in the present study.”
L205-206: Please rephrase
L206-209: Please rephrase
L267-269: Please rephrase (see comment L50-54)
Author Response
L26-27: “Rubber seed kernel does not contain sufficient levels of crude protein to serve as its primary source in ruminant diets.”
-L26-27: Already changed, please see in text.
L30: “…enhanced levels of crude protein…”
-L30: Already changed, please see in text.
L31: “provided” instead of “fed”
-L31: Already changed, please see in text.
L31-32: Please delete “The impact of hematological, microbial protein synthesis, milk production, and milk quality in dairy cows is undefined.”
-Already deleted, please see in text.
L34: “feed intake” instead of “the consumption of feed”
-L33: Already changed, please see in text.
L35-36: “Hence, yeast-fermented rubber seed kernel could be used as a source of protein in diets of lactating dairy cows.”
-L33-35: Already changed, please see in text.
L49-50: Please delete “at 10% in concentrate”
-Already deleted, please see in text.
L50-54: This conclusion is not supported by your findings. P-linear shows a linear trend after increasing YERSEK levels. As shown, the level of both 10 and 20% resulted in a decrease of milk fat and total solid.
-L48-51: Already changed to “In conclusion, the use of YERSEK in concentrate diet could be used as a protein source without negative effects on feed intake, digestibility, hematology, microbial protein synthesis and milk yield. However, it reduced milk fat and total solids of tropical lactating dairy cows.
L89: “yield”
-L86: Already changed, please see in text.
L98-99: “The yeast was activated by mixing 5 g of S. cerevisiae with 100 ml of water, and 20 g of sucrose for an hour at room temperature.”
-L95-96: Already changed, please see in text.
L103-104: “and determining their chemical composition” instead of “and having their chemical 103 makeup examined”
-L100-101: Already changed, please see in text.
L126: “Patient”?
-L123: Already changed to “Urination was induced by strokes on the side of the vulva.”, please see in text.
L203: “contrast” instead of “disagreement with the current study”
-L200: Already changed, please see in text.
L205: “…of dairy cows in the present study.”
-L201: Already changed, please see in text.
L205-206: Please rephrase
-L202-203: Already changed to “……which could be suitable levels for rumen microbial activity and also nutrient intake”, please see in text.
L206-209: Please rephrase
-Already deleted, please see in text.
L267-269: Please rephrase (see comment L50-54)
-L259-262: Already changed to “The inclusion of YERSEK at 10%-20% in concentrate could be used as a protein source and had no effect on feed utilization, hematological parameters, microbial protein synthesis and milk production, while milk fat and total solids were decreased in tropical lactating dairy cows.”, please see in text.
Reviewer 3 Report
The paper has been improved, but the following points must be checked:
Line 89; check “milk yeild and milk composition in tropical lactating dairy cows.”
Line 99-100: Check 'Urea, molasses and water were combined in a ratio of 40:42:100 to create the liquid medium'. Is the ratio 40:42:100 correct?
Line 130: Milk samples were also collected (manually or mechanically? Specify).
Line 150 and throughout text: check '%DM', rewrite as '% DM'.
Line 151 and throughout text: check 'MJ/kgDM ', rewrite as “MJ/kg DM”.
Line 178: check “lactose, , and”
Line 310: 'Dairy heifers'. Fermentation 2022, 8, 288." Year of publication must be written in bold, check all references.
Author Response
The paper has been improved, but the following points must be checked:
Line 89; check “milk yeild and milk composition in tropical lactating dairy cows.”
-L86: Already changed, please see in text.
Line 99-100: Check 'Urea, molasses and water were combined in a ratio of 40:42:100 to create the liquid medium'. Is the ratio 40:42:100 correct?
-L96-97: Already changed to “To create the liquid medium, combine 20 g of molasses, 100 ml of distilled water, and 42 g of urea.”, please see in text.
Line 130: Milk samples were also collected (manually or mechanically? Specify).
-L127-128: Already changed to “Milk samples were also collected by milking machines…..”, please see in text.
Line 150 and throughout text: check '%DM', rewrite as '% DM'.
-L147, 148 and 150: Already changed, please see in text.
Line 151 and throughout text: check 'MJ/kgDM ', rewrite as “MJ/kg DM”.
-L148-151: Already changed, please see in text.
Line 178: check “lactose, , and”
L175: Already changed, please see in text.
Line 310: 'Dairy heifers'. Fermentation 2022, 8, 288." Year of publication must be written in bold, check all references.
-L302: Already changed and checked all references, please see in text.